# Relationships between Regeneration of Qinghai Spruce Seedlings and Soil Stoichiometry across Elevations in a Forest in North-Western China

**Xiurong Wu** [1,2], **Peifang Chong** [1,*], **Erwen Xu** [2], **Weijun Zhao** [2], **Wenmao Jing** [2], **Ming Jin** [3], **Jingzhong Zhao** [2], **Shunli Wang** [2], **Rongxin Wang** [2] and **Xuee Ma** [2]

[1] College of Forestry, Gansu Agricultural University, Lanzhou 730000, China; xrwu2013@163.com
[2] Academy of Water Resource Conservation Forests of Qilian Mountains in Gansu Province, Zhangye 734000, China; xuerweng@126.com (E.X.); zhaoweijun1019@126.com (W.Z.); maodanjing@126.com (W.J.); zhaojz_5659@163.com (J.Z.); wangshun123_78@163.com (S.W.); zywangrx@163.com (R.W.); maxuee111@126.com (X.M.)
[3] College of Qilian Mountain Ecological Research, Hexi University, Zhangye 734000, China; shyjinming@163.com
[*] Correspondence: zhongpf@gsau.edu.cn; Tel.: +86-0931-7631200

**Abstract:** Qinghai spruce (*Picea crassifolia* Kom.) is an ecologically important species in the forest ecosystem of the Qilian Mountains region in China. Natural regeneration of this species is critical to maintaining forest ecosystem function. Here, we analyzed several biological indicators among naturally regenerating Qinghai spruce across several elevations in the Pailugou watershed. Specifically, seedling density, basal diameter (BD), and plant height were measured, as were soil physicochemical parameters, at 2700 m, 3000 m, and 3300 m above sea level. Differences in the regeneration indicators and correlations between the indicators and soil parameters were then assessed across elevations. The results showed that soil stoichiometry was more sensitive to changes in elevation than seedling indicators were. Furthermore, seedling density was positively correlated with soil pH, whereas BD was positively correlated with the carbon-to-nitrogen ratio (C/N), the carbon-to-phosphorus ratio (C/P), and soil organic carbon (SOC) contents. None of the analyzed soil stoichiometry parameters had a significant impact on elevation-specific differences in seedling density. However, soil pH, SOC, and C/N significantly affected variations in seedling basal diameter at different elevations. Finally, soil pH, SOC, C/N, and the carbon-to-phosphorus ratio significantly affected variations in seedlings' heights at different elevations. This study provides a strong theoretical basis for further understanding of the mechanisms associated with Qinghai spruce regeneration, ultimately contributing to rational protection and management strategies for this important natural resource.

**Keywords:** *Picea crassifolia*; natural regeneration; soil stoichiometry; regeneration characteristics; elevation gradient



## 1. Introduction

Forest ecosystems are the most structurally complex, widespread, and biologically diverse ecosystems on land. They not only provide shelter and habitat for many species, but also play important roles in climate mitigation and disaster resilience [1,2]. Soil is the most extensive organic matter pool in the terrestrial biosphere and, therefore, acts as the prime agent for material cycling in terrestrial ecosystems. It provides plants with essential nutrients for sustenance, growth, and development [3,4]. Soil stoichiometry, as an indicator of soil condition, is influenced by climate, moisture, litter input, and elevations, with elevations having a more significant effect on soil stoichiometric characteristics [5]. Different soil stoichiometric characteristics can directly affect plant growth and development and distribution characteristics [6]. In montane forest, alterations in environmental variables,

primarily manifested as shifts in vertical gradients, exert an influence on the distribution and stoichiometry of plant nutrients. Consequently, in montane forest, variations in environmental variables arising from shifts in vertical gradients play a crucial role in the regeneration and distribution of species [7]. For example, in the study of soil stoichiometric characteristics of Dongting Lake, it was found that the soil total nitrogen, soil organic carbon, carbon-to-nitrogen ratio, carbon-to-phosphorus ratio, and nitrogen-to-phosphorus ratio increased with elevation, whereas the change in total phosphorus (TP) did not change significantly [8]. Differences in elevation-related soil organic carbon (SOC) significantly impact the distribution of sweet chin-kapin (*Toona sinensis* Roem.) [9]. The most important environmental factors affecting regeneration of tropical secondary forests are the soil nutrient status and thickness of dead wood [10].

Qinghai spruce (*Picea crassifolia* Kom.), belonging to the *Pinaceae* family and the *Picea* genus, is the most important structural and dominant species in the Qilian Mountains Forest, accounting for 79.6% of the total forest area in the Qilian Mountains. It is mainly distributed as a pure forest and is an indigenous evergreen coniferous tree growing at elevations of 2500 to 3300 m. The growth morphology is illustrated in Figure 1. It plays an important role in windbreak, sand fixation, water conservation, and local climate regulation [11]. Deforestation, unsustainable land resource utilization, and inadequate management of Qinghai spruce forests since the 1960s have resulted in poor regeneration of Qinghai spruce. In 2001, China introduced a series of policy documents that mandated the cessation of all commercial natural forest logging activities in the Qilian Mountains region. In recent years, the Qilian Mountains National Nature Reserve has achieved noteworthy success in natural forest protection projects [12]. However, the site is located in an ecologically fragile area with delicate natural conditions, a variable climate, and a monolithic forest structure. Structural integrity is extremely difficult to restore after it is compromised [13]. Research on the regeneration of Qinghai spruce primarily centers on its spatial characteristics and influencing mechanisms. The majority of studies concentrate on identifying factors that impact the regeneration of Qinghai spruce [14,15]. Several studies have investigated the impacts of soil moisture and fertility on Qinghai spruce growth and development. For example, increased temperature and humidity facilitate earlier spruce growth, and a higher altitude postpones budding [16]. The carbon density in the tree layer has been shown to decrease along with increasing elevation, whereas the carbon density in the soil layer first decreases, then increases. Furthermore, the percentage of soil carbon density out of the total carbon density increases across elevations [15]. There are significant positive correlations of elevation with soil organic carbon content and carbon density in the Qinghai spruce tree layer, and a significant negative correlation between elevation and the mean summer air temperature [17]. However, links between the natural regeneration features of Qinghai spruce and soil stoichiometric properties at varying elevations remain unexplored.

To address this gap in knowledge, we examined the associations of three Qinghai spruce regeneration indicators (density, basal diameter (BD), and plant height) with elevation and soil stoichiometric characteristics. The following hypotheses were addressed: (1) Qinghai spruce regeneration and soil stoichiometry would differ based on elevation, (2) regeneration density, BD, and plant height would show specific associations with soil stoichiometric parameters at each elevation, and (3) variations in soil stoichiometry would affect regeneration density, BD, and plant height across various altitudinal gradients. This study was designed to provide new insights into the relationships across elevations, soil characteristics, and Qinghai spruce regeneration, providing valuable information for protection of this ecologically important species.

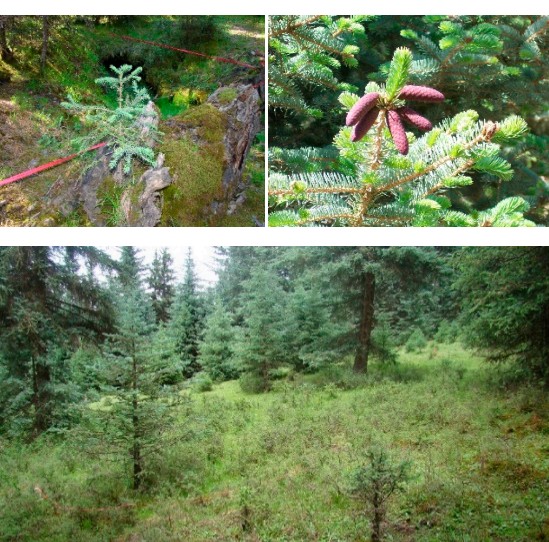

**Figure 1.** The growth stages of Qinghai spruce, from a seedling to an adult tree.

## 2. Materials and Methods

### 2.1. Study Area

The study area was located in the Pailugou watershed in the Qilian Mountains National Park (100°17′0″–100°18′30″ E, 38°32′0″–38°33′30″ N). The mean annual temperature was 5.4 °C, ranging from a monthly average high of 12.2 °C in July to a monthly average low of −12.9 °C in January [15]. The region has a continental alpine mountain climate. The average annual precipitation is 416 mm, with an annual potential evaporation of 1011.3 mm and an annual sunshine duration of 1562.6 h [18,19]. There is an intricate topography with significant variations in elevation. The vegetation types are montane grassland, montane forest-steppe, subalpine scrub-meadow, and alpine snow and ice vegetation. The soil types vary by elevation and comprise, from lowest to highest elevation, mountain chestnut-calcium, mountain greyish-brown, alpine meadow, and alpine desert soil. The primary tree species dominating the area are Qinghai spruce and *Sabina przewalskii* Kom., and the most abundant shrubs are *Caragana jubata* Poir., *Potentilla fruticosa* L., *Salix gilashanica* C., *and Potentilla glabra* Lodd. The predominant herbaceous plants in the area are *Carex atrata* L., *Achnatherum splendens* Trin., *Potentilla chinensis* Ser., and *Polygonum viviparum* L.

### 2.2. Experimental Design

Based on the life history characteristics of Qinghai spruce, a population size classification was performed [20,21]. Seedlings with a height of <2 m and a diameter at breast height (DBH) of <50 mm were categorized as regeneration seedlings. In this context, the quantification of seedling diameter at breast height (DBH) is unfeasible; therefore, the alternative is to measure the basal diameter (BD) instead. In 2022, experimental sample plots were selected at 2700 m (low elevations), 3000 m (mid elevations), and 3300 m (high elevations) above sea level (a.s.l.) (Figure 2). The selected sample site is a pure Qinghai spruce forest, which originated from natural secondary forest and is now a middle-aged forest. The primary human activity in the area is grazing. The soil type in each plot was a mountainous gray cinnamon soil, and the texture was considered loam, with a sand content of 34%–58% [22]. The soil thickness was 50–80 cm. Six 20 × 20 m sample plots were selected at different elevations, and the soil depth, aspect, and slope were recorded (Table 1). Each sample plot was partitioned into 16 small squares of 5 × 5 m, 3 of which were selected at random for measurements of Qinghai spruce seedling density, basal diameter (BD), and plant height. Regeneration density was calculated from the number of seedlings within each sample plot, whereas seedling height and basal diameter were measured with a measuring tape and a caliper, respectively. Basic information concerning the elevation, soil depth, aspect, and slope of each plot was also recorded during field work (Table 1).

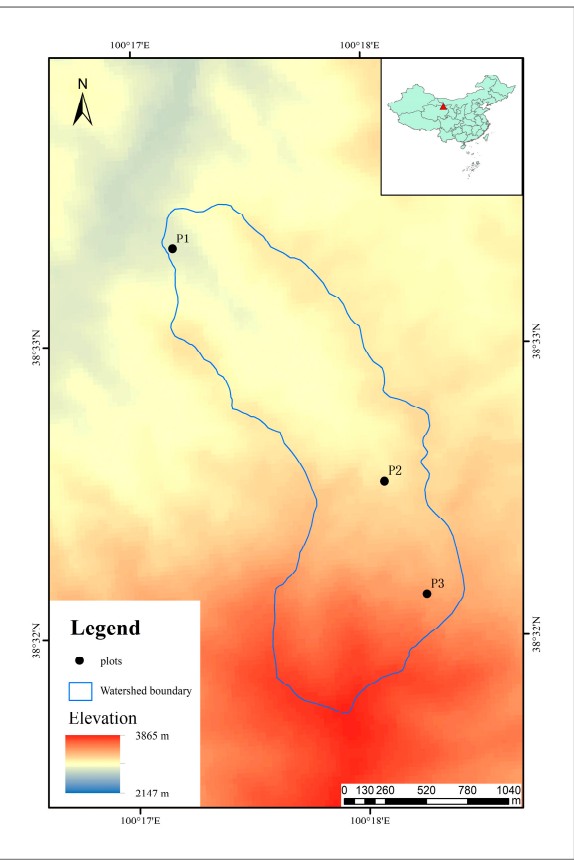

**Figure 2.** Digital elevation model (DEM) of the study area showing the locations of Qinghai spruce plots. P1–P3 represent the locations of three plots at elevations of 2700 m, 3000 m, and 3300 m above sea level, respectively.

**Table 1.** Basic characteristics of the three Qinghai spruce regeneration seedlings plots (mean value ± standard deviation).

| Plots | Coordinates | Elevation (m above Sea Level) | Soil Thickness (cm) | Aspect (°) | Slope (°) | Seedlings Density (Trees·ha$^{-1}$) | Average Basal Diameter (mm) | Average Seedlings Height (m) |
|---|---|---|---|---|---|---|---|---|
| P1 | 100°17′10.3″ E 38°33′19.9″ N | 2700 | 70 ± 15 | NE | 27~33 | 7267 ± 6436 | 19.39 ± 5.50 | 0.81 ± 0.31 |
| P2 | 100°18′4.8″ E 38°32′31.8″ N | 3000 | 60 ± 12 | NE | 25~36 | 41,333 ± 2505 | 12.91 ± 3.89 | 0.52 ± 0.26 |
| P3 | 100°18′15.5″ E 38°32′8.5″ N | 3300 | 50 ± 8 | NE | 28~40 | 9044 ± 7280 | 15.05 ± 3.29 | 0.64 ± 0.22 |

Soil was collected from the four corners and center of each plot at a depth of 0–10 cm and uniformly mixed to form a single sample per plot. These served as representative samples of the overall soil composition. Soil moisture content (SMC) was measured using the drying method, soil pH was determined using the leaching electrode method, SOC was analyzed with the potassium dichromate method, TN was quantified using a Kjeldahl nitrogen meter, and total phosphorus (TP) was assessed using the acid solubilization-molybdenum antimony anti-stibnite colorimetric method [22].

### 2.3. Data Processing

Normality was assessed for each indicator (density, BD, and height) with the Shapiro test. Differences in density, BD, plant height, and soil stoichiometric attributes across elevations were assessed with one-way analysis of variance (ANOVA). Multiple comparisons were conducted with Dunn's test, followed by detrended correspondence analysis (DCA),

which revealed a first axis result of <3. Redundancy analysis (RDA) was, therefore, selected to examine correlations between the three seedling indicators and soil stoichiometric properties. Adonis multivariate ANOVA was utilized to examine differences in seedling indicators based on soil physicochemical characteristics at different elevations. The distance matrix was generated using Bray–Curtis distance, and the Monte-Carlo permutation test was set to 999. Data were recorded and initially collated with Microsoft Excel, version 2019 (Microsoft Corporation, Redmond, WA, USA); Normality tests and ANOVA were conducted in Origin Pro, version 2022SR1 (OriginLab Corporation, Northampton, MA, USA); RDA was performed with Canoco5, version 5.15 (Microcomputer Power, Ithaca, NY, USA); and Adonis multivariate ANOVA was conducted in RStudio, version 4.1.3. (RStudio Inc., Boston, MA, USA).

## 3. Results

### 3.1. Regeneration Characteristics of Qinghai Spruce at Varying Elevations

We first compared the regeneration seedling density, BD, and plant height across elevations (Figure 3). Generally, the seedling density was higher and more scattered at 2700 m, but lower and more concentrated at 3000 m. There were significant differences in the regeneration seedling density at 3000 m compared to 3300 m ($p < 0.05$). Furthermore, the regeneration seedling density was significantly higher at a single 3300 m plot than at other plots at the same elevation. This was likely due to the relatively gentle slope of this plot, which could have had a concentrating effect on Qinghai spruce seedlings. Overall, the seedling BD was highest at 2700 m, with a more dispersed distribution than at the other elevations. There were significant differences between altitudes: seedlings at 3000 m had a lower average diameter than those at 3300 m. The latter exhibited diameters ranging from those found at 2700 m to those found at 3000 m, with a concentrated distribution. The difference in BD between seedlings regenerated at 2700 m and those regenerated at 3000 m was highly significant ($p < 0.01$). Seedlings at 2700 m exhibited a greater overall height with a more dispersed distribution, while those at 3000 m had a lower overall height with a more centralized distribution. There were significant differences in height between seedlings at 2700 m compared to 3000 m ($p < 0.01$). Seedlings at 2700 m were taller but had a more dispersed height distribution, whereas seedlings at 3000 m were shorter with a more distributed height ($p < 0.01$). Seedlings in one plot at 3000 m were significantly taller than those from other plots at the same elevation. However, this finding was not distinctive and was sustained. Overall, the three regeneration indicators decreased among seedlings at 3000 m compared to 2700 m, then increased again at 3300 m.

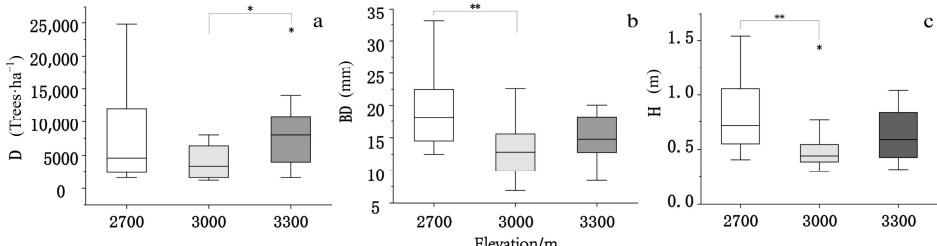

**Figure 3.** Distribution of seedling indicator values across elevations. Seedlings were measured to assess (**a**) density (D), expressed in Trees·ha$^{-1}$, (**b**) basal diameter (BD), expressed in mm, and (**c**) plant height (H), expressed in m. * $p < 0.05$ and ** $p < 0.01$ (one-way analysis of variance).

### 3.2. Characteristics of Soil Stoichiometry at Different Elevations

We next examined variations in soil nutrients in the Qinghai spruce forest across the three elevations (Figure 4). With the exceptions of soil pH and TN, all the soil parameters first decreased, then increased along with the altitude. SMC, TN, C/P, N/P, and SOC peaked at 2700 m and reached minima at 3000 m. C/N also exhibited the maximum value at 2700 m but remained virtually unchanged at 3000 m and 3300 m. The pH was highest at

3000 m and lowest at 2700 m. In contrast, TN was lowest at 3300 m and highest at 2700 m. The differences in soil pH, TN, C/P, N/P, SOC, and SMC were significant between 2700 m and 3000 m and between 3000 m and 3300 m ($p < 0.05$); in contrast, the differences between 2700 m and 3300 m were not statistically significant. There were no significant differences in C/N or TP between elevations.

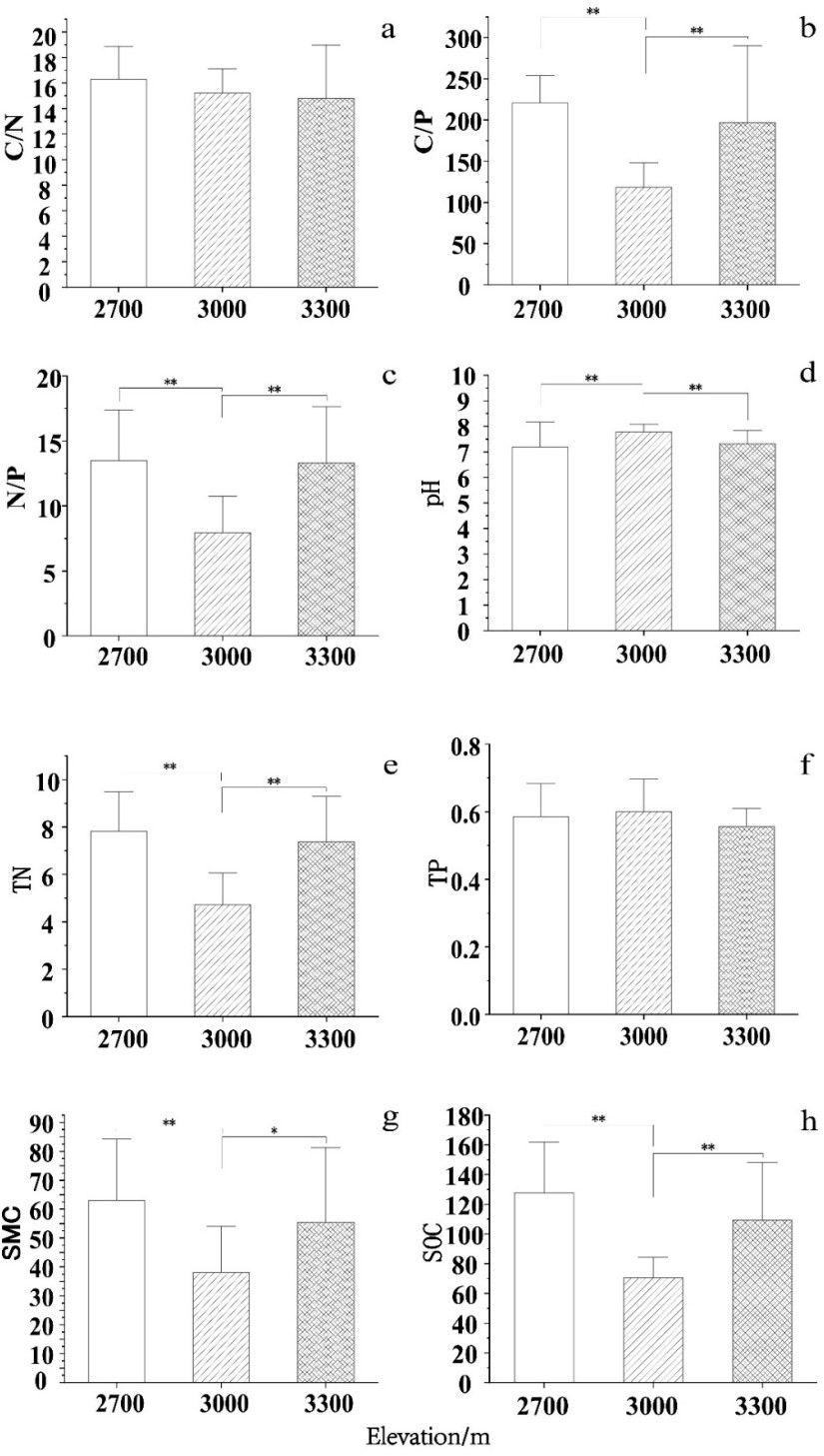

**Figure 4.** Soil stoichiometric characteristics across elevations (values are means ± SE). Soil samples were assessed to determine the (**a**) carbon-to-nitrogen ratio (C/N), (**b**) carbon-to-phosphorus ratio (C/P), (**c**) nitrogen-to-phosphorus ratio (N/P), and (**d**) pH. (**e**) Total nitrogen (TN) and (**f**) total phosphorus were expressed in g kg$^{-1}$, (**g**) soil moisture content (SMC) was expressed in %, and (**h**) soil organic carbon (SOC) was expressed in g kg$^{-1}$. * $p < 0.05$ and ** $p < 0.01$.

### 3.3. Relationships between Regeneration Indicators and Soil Physicochemical Characteristics

RDAs were next conducted for seedling density, BD, and plant height with the analyzed soil parameters (Figure 5). At 2700 m, the eight soil parameters accounted for 52.3% of soil variability: the first two axes accounted for 33.4% and 11.9%, respectively. The angles indicated that seedling density was positively correlated with pH and TP, and negatively correlated with C/N, C/P, N/P, SOC, and SMC. BD was positively correlated with TP, TN, C/N, C/P, SOC, and SMC, and plant height was positively correlated with TN, TP, C/N, SOC, and C/P, and negatively correlated with pH and N/P. At 3000 m, the eight soil parameters accounted for 64.9% of the soil variance, with the first two axes explaining 48.5% and 15.1% of the variance, respectively. Seedling density was positively correlated with the pH and TP, and negatively correlated with the TN, C/N, C/P, N/P, SOC, and SMC. Furthermore, BD was positively correlated with C/N, C/P, SOC, and SMC, and negatively correlated with TN, TP, N/P, and pH. Plant height was positively correlated with C/N, C/P, N/P, SOC, and SMC, and negatively correlated with TN, TP, N/P, and pH. At 3300 m, seedling density was positively correlated with pH, TP, TN, and N/P, and negatively correlated with C/N. BD was positively associated with C/N, C/P, N/P, TN, TP, SOC, and SMC. Plant height was positively correlated with C/N, C/P, N/P, TN, TP, SOC, and SMC, and negatively correlated with pH.

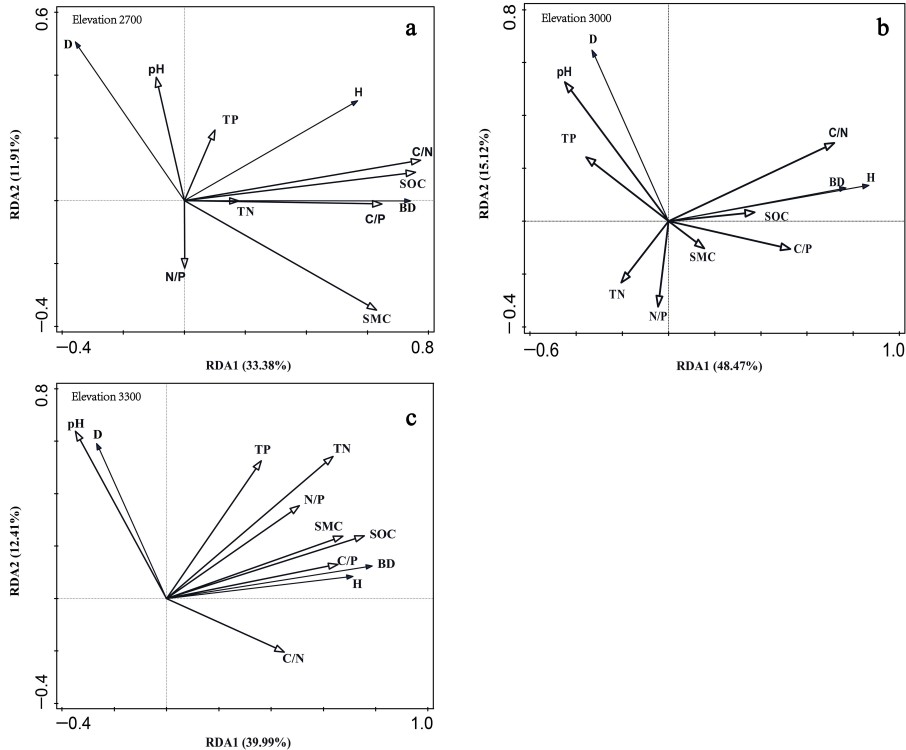

**Figure 5.** Redundancy analysis of regeneration indicators—soil stoichiometric parameters: density (D), basal diameter (BD), and plant height (H), and soil stoichiometry, at three elevations. Samples were collected at (**a**) 2700 m, (**b**) 3000 m, and (**c**) 3300 m. pH, pH value; TP, total phosphorus; TN, total nitrogen; N/P, nitrogen-to-phosphorus ratio; SMC, soil moisture content; SOC, soil organic carbon; C/P, carbon-to-phosphorus ratio; C/N, carbon-to-nitrogen ratio.

RDA was also used to rank the contribution rates of soil parameters to soil variability at different elevations (Figure 6). At 2700 m, C/N contributed 40.3% to the observed variations in soil stoichiometric characteristics ($p < 0.05$), whereas the other indicators showed no significant contributions. At 3000 m, C/N also contributed 40.6% to variations in soil stoichiometric characteristics; additionally, N/P contributed 25.5% to variations in soil stoichiometric characteristics ($p < 0.05$). At 3300 m, SOC was responsible for 44.3% of

the variation in soil stoichiometric characteristics ($p < 0.05$), whereas the other indicators showed no significant contributions.

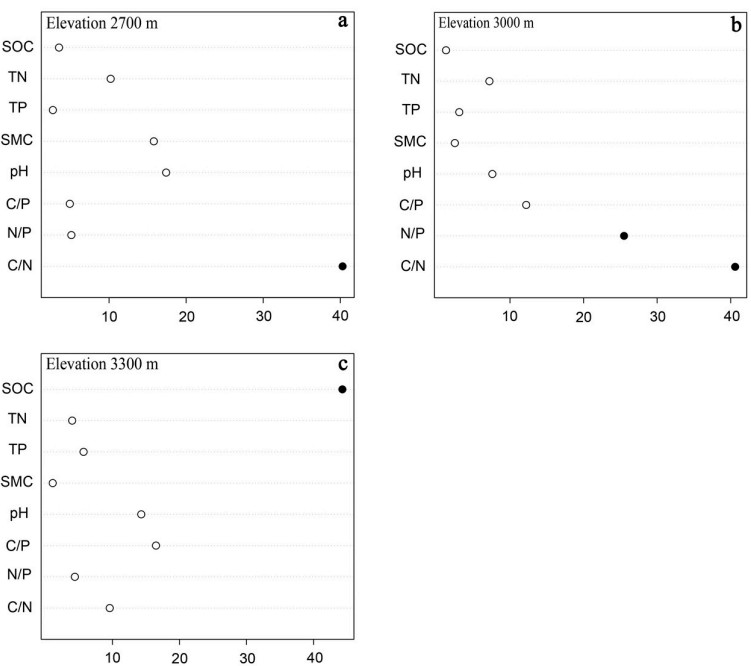

**Figure 6.** Analysis of soil chemical stoichiometric parameter variability at different elevations based on the redundancy analysis (RDA) model. Samples were collected at elevations of (**a**) 2700 m, (**b**) 3000 m, and (**c**) 3300 m. Regeneration indicators were seedling density, basal diameter, and height. pH, pH value; TP, total phosphorus; TN, total nitrogen; N/P, nitrogen-to-phosphorus ratio; SMC, soil moisture content; SOC, soil organic carbon; C/P, carbon-to-phosphorus ratio; C/N, carbon-to-nitrogen ratio.

For the regeneration indicators categorized based on different elevations, through Adonis analysis, $R^2$ was employed to illustrate the extent of disparity in the indicators among the different elevation groups. None of the soil variables had significant impacts on seedling density at different elevations (Table 2). However, pH, SOC, and C/N had significant effects on BD across elevations, with $R^2$ values of 0.161, 0.245, and 0.0690, respectively. Of these three variables, SOC had the greatest effect and C/N had the smallest. There were also significant effects of pH, SOC, C/N, and C/P on plant height ($R^2 = 0.117$, 0.201, 0.134, and 0.08, respectively). Here again, SOC had the greatest effect, whereas C/P had the smallest effect. Overall, the effects of SOC were strongest and the effects of C/N on spruce regeneration indicators were the weakest.

**Table 2.** Effects of soil stoichiometry at different elevations on seedling regeneration indicators.

| Variables | Density | | Basal Diameter (BD) | | Height | |
|---|---|---|---|---|---|---|
| | $R^2$ | F | $R^2$ | F | $R^2$ | F |
| pH | 0.006 | 0.344 | 0.161 ** | 14.972 | 0.117 ** | 11.455 |
| SOC/g kg$^{-1}$ | 0.026 | 1.432 | 0.245 ** | 22.766 | 0.201 * | 19.671 |
| TN/g kg$^{-1}$ | 0.054 | 2.954 | 0.014 | 1.315 | 0.006 | 0.571 |
| TP/g kg$^{-1}$ | 0.017 | 0.912 | 0.001 | 0.068 | 0.001 | 0.054 |
| C/N | 0.007 | 0.393 | 0.069 * | 6.360 | 0.134 ** | 13.093 |
| C/P | 0.027 | 1.476 | 0.023 | 2.148 | 0.08 * | 7.810 |
| N/P | 0.005 | 0.279 | 0.001 | 0.087 | 0.002 | 0.213 |
| SMC/(%) | 0.037 | 2.047 | 0.001 | 0.083 | 0.001 | 0.081 |

pH, pH value; SOC, soil organic carbon; TN, total nitrogen; TP, total phosphorus; C/N, carbon-to-nitrogen ratio; C/P, carbon-to-phosphorus ratio; N/P, nitrogen-to-phosphorus ratio; SMC, soil moisture content. * $p < 0.05$ and ** $p < 0.01$.

## 4. Discussion

### 4.1. Soil Stoichiometry and Regeneration Indicators Correlated with Qinghai Spruce across Elevations

Elevation impacts the temperature, light, and moisture of a forest environment. These factors, in turn, affect the stand structure, plant distribution, and species diversity. As a result, there is extensive spatial variation in the vegetation of a given community, as indicated by various biological factors and the physicochemical characteristics of the understory soil [23,24]. For example, elevation influences *Tetracentron sinense* Oliv. distribution by affecting environmental factors, including temperature and moisture. This reflects the plant's ecological characteristics and environmental responses [25]. *Pinus tabulaeformis* Carr. forests show differing responses to the environment depending on the elevation, where increases in altitude are associated with gradual increases in density [26]. Elevation determines numerous ecosystem properties and processes in the mountains of subtropical China, with changes in elevation strongly impacting soil stoichiometry [27]. The present study highlighted variations in Qinghai spruce regeneration indicators and soil stoichiometry based on elevation shifts. Specifically, we identified significant differences between seedling density at 3000 m and 3300 m during regeneration. In addition, there were significant differences in both BD and plant height between 2700 m and 3000 m. However, soil stoichiometric properties were more sensitive than plant indicators were to elevations, and there were statistically significant differences in C/P, N/P, pH, TN, SOC, and SMC between 2700 m and 3000 m and between 3000 m and 3300 m.

Previous studies have shown that soil microbial activity has a direct effect on the rate of humus decomposition—the hydrothermal variations resulting from differences in elevation directly impact soil stoichiometric properties [28]. Meanwhile, plants possess the ability to self-regulate and adjust their growth strategies [29]. Consequently, the impact of elevation differences on seedlings is relatively minor compared to the profound effects on soil stoichiometric properties [30]. In general, Qinghai spruce regeneration seedlings exhibit a pattern of decreasing and then increasing density, basal diameter, and plant height with rising elevation. This mirrors the trends observed in the soil total nitrogen (TN), soil moisture content (SMC), soil organic carbon (SOC), carbon-to-phosphorus ratio (C/P), and nitrogen-to-phosphorus ratio (N/P) across elevations. As elevation increases, temperature decreases, and it is possible that favorable hydrothermal conditions at 2700 m result in higher stand biomass, increased material transported to the soil by plants, enhanced soil microbial activity, and a faster humus decomposition rate. Consequently, this leads to an improved regeneration status and elevated soil stoichiometry [31]. At an elevation of 3000 m, the hydrothermal conditions are less favorable, resulting in lower forest stand biomass. Larger age classes of spruce dominate intraspecies competition, leading to a poorer regeneration status and lower soil stoichiometry. At an elevation of 3300 m, the hydrothermal conditions are relatively worse, with reduced soil microbial activity and a slower humus decomposition rate. However, a significant reduction in older spruce and weakened intraspecific competition contribute to an improvement in the regeneration status and soil stoichiometry [32].

### 4.2. Relationships between Qinghai Spruce Regeneration and Soil Stoichiometry

Forest regeneration is impacted by various factors, including species-specific biology, faunal behavior, microhabitat characteristics, and the abundance of older plants. Soil stoichiometry slightly influences regenerating seedling growth, development, and distribution patterns [33,34]. A prior study on the renewal characteristics and influencing factors in mixed forests of the Qinling Mountains found that SMC is the primary factor influencing the distribution of oil pine seedlings, and that SOC and quick-acting phosphorus have the most significant impacts on Hua Shan pine seedlings [35]. Another study on soil nutrients' impacts on spruce secondary forest regeneration discovered a significant positive correlation between soil pH and the number of spruce seedlings [36]. These prior results are comparable to the findings of the present study. Here, we found a positive correlation

between Qinghai spruce seedling density and pH and a negative correlation with C/N at all three elevations. This suggested that indigenous Qinghai spruce regeneration favors either neutral or slightly alkaline soils. This preference may have been influenced by micro-topographical factors. For example, on gentle slopes where organic matter and plant seeds accumulate [37], soil with sufficient aeration and microbial activity supports good soil structure, in tandem with the biogenic root system. This combination makes it harder for salty ions to be washed away by water flow and for a higher pH to form [38].

Qinghai spruce regeneration may have varied between elevations due to environmental changes [39]. Regeneration seedlings' BD and plant height also varied with elevation, and there was a slight elevation-based difference in the relationship between the number and degree of soil stoichiometric indicators. However, the overall performance demonstrated a positive correlation with the C/N, C/P, and SOC. Specifically, the C/N significantly impacted seedlings at 2700 m and 3000 m, the N/P had a notable effect on seedlings at 3000 m, and SOC significantly influenced seedlings at 3300 m. SOC serves as a source of nutrients for plant-associated microorganisms and enhances soil physicochemical and biological properties by influencing the soil bulk density, increasing porosity, promoting the soil aggregation structure, and enhancing soil water absorption, water holding, and aeration [40,41]. Thus, SOC plays a crucial role in plant growth, particularly under adverse environmental conditions, but also when conditions are generally conducive to plant growth. Furthermore, the soil C/N and C/P statuses offer valuable insights into the behavior of soil microorganisms, and microbial decomposition of humus matter improves plant nutrients and stimulates growth. Additionally, the processes of plant growth and decay contribute materials and energy to the soil, highlighting the pivotal roles of soil microorganisms in promoting plant succession [42,43].

Plant regeneration strategies and the factors that influence biological indicators can differ between sites even at the same elevation due to variations in topography, hydrothermal conditions, and other factors, which result in different nutrient inputs to the soil and cause variations in soil stoichiometry [44]. Although here pH was correlated with the regenerated Qinghai spruce seedling density at various altitudes, we found no significant impacts of soil parameters on seedling density at any elevation. However, soil pH, SOC, and C/N significantly influenced elevation-specific differences in BD. Additionally, soil pH, SOC, C/N, and C/P all had significant impacts on differences in plant height between elevations. Overall, Qinghai spruce seedling regeneration is impacted by a range of factors beyond soil stoichiometric properties. Previously identified factors of this nature include the distance from cohorts of older plants, surface temperature, light exposure, slope orientation and inclination, anthropogenic activities, and competition with other plant types, such as shrubs [45–48]. Light conditions and the forest window environment are also known to strongly affect the regeneration density [28,49]. This explains the relatively small role of soil variations in affecting the regeneration density at each elevation. The correlations between seedling growth (BD and height) and soil nutrients are predominantly linked to density [50]. Nutrient absorption directly from the soil is essential for plant growth and development. Thus, differing levels of soil nutrients can have a significant impact on seedling growth, affecting both BD and height to produce diverse growth patterns. This consequently leads to differences in seedling BD and height between elevations.

## 5. Conclusions

The regeneration indicators and soil chemical stoichiometric characteristics of Qinghai spruce followed a consistent trend along the altitude gradient, initially decreasing and then increasing. Notably, there were marked differences between the regeneration indicators and most soil chemical stoichiometric parameters of Qinghai spruce across elevations: pH, SOC, and C/N had significant effects on basal diameter (BD) and plant height across elevations, while other parameters had a relatively small impact. The sensitivity of soil chemical stoichiometric characteristics to elevation changes surpassed that of regeneration indicators. This is because hydrothermal changes due to elevation differences directly

affect the chemical stoichiometry characteristics, whereas Qinghai spruce have the ability to self-regulate and also adjust their own growth strategies.

The correlation between seedling density and soil pH was positive, while basal diameter and plant height showed positive correlations with the C/N, C/P, and SOC. Elevation-induced variations in soil stoichiometry did not significantly impact seedling density differences. However, soil pH, SOC, and C/N significantly influenced changes in the seedling diameter at each elevation. Changes in the seedling height at each elevation were significantly affected by soil pH, SOC, C/N, and C/P.

Overall, this study advances our comprehension of the population regeneration distribution of Qinghai spruce in the Qilian Mountains and its correlation with soil chemical stoichiometry characteristics. Despite the stringent forest management and the inherent soil characteristics of the Qilian Mountains, manual intervention remains a challenge. Nevertheless, artificial forest regeneration of Qinghai spruce in the Qilian Mountains can effectively regulate soil chemical elements, promoting its sustainable regeneration.

**Author Contributions:** Conceptualization, X.W. and P.C.; methodology, W.Z., S.W. and M.J.; software, W.J. and R.W.; investigation, X.W., W.Z., J.Z. and X.M.; data curation, W.Z.; writing—original draft preparation, X.W.; writing—review and editing, X.W., P.C. and M.J.; supervision, E.X.; project administration, E.X. All authors have read and agreed to the published version of the manuscript.

**Funding:** This work was financially supported by the National Natural Science Foundation of China (U21A20468 and 32060247), the Natural Science Foundation of Gansu Province (22JR5RG1030 and 22JR5RG1029), and the central government of Gansu Province guides local science and technology development funds (22ZY2QG001).

**Data Availability Statement:** Data are contained within the article.

**Conflicts of Interest:** The authors declare no conflicts of interest.

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
