# Peer review of "Relationships between Regeneration of Qinghai Spruce Seedlings and Soil Stoichiometry across Elevations in a Forest in North-Western China"

_forests, doi:10.3390/f15020267_

Round 1

Reviewer 1 Report

Comments and Suggestions for Authors

The soil nutrient stocks consisting of the stocks of N, P and Soil organic carbon and the stoichiometry are key indicators of soil quality of an ecosystem. This manuscript is providing a detailed insight on the relationship between regenerating Qinghai spruce seedlings and soil stoichiometry across elevations in a forest in China. The findings of this research may be helpful in understanding the regeneration behavior of the species in response to the soil nutrient dynamics of the area. The objectives and findings of this study hold novel scientific approach and all the sections of the manuscript are very well written. 

1. Line no. 75: Please give full form of LAMs

2. In materials and methods section:

The methods adopted for determining the density, basal diameter and plant height should also be described. 

Also mention the units of SOC, TN and TP

3. In result section 

Line no. 210: Write Nitrogen instead of ni-trogen  

Table 2: Write Height instead of Hight

4. In Discussion 

Line no. 247 to 248 - Reference should be given for the statement 

Line no. 287: Write between instead of be-tween 

Comments on the Quality of English Language

Minor typographical errors in the manuscript needs correction. 

Author Response

#comment-1:

  1. Line no. 75: Please give full form of LAMs

Response 1: Thanks for your suggestion, LAMs stands for Local Abundance Models, the initial aim was to convey the alterations in various forest regions, but our study focuses on a single, pure forest, we consider it inappropriate to use this word and will give up using it in manuscript.

Thanks again for your valuable suggestion.

#comment-2:

  1. In materials and methods section: The methods adopted for determining the density, basal diameter and plant height should also be described. Also mention the units of SOC, TN and TP

Response 2: Thanks for your suggestion. we have added the methods adopted for determining the density, basal diameter and plant height to the manuscript in line 126-128.

Thanks again for your valuable suggestion.

#comment-3:

  1. In result section:Line no. 210: Write Nitrogen instead of ni-trogen, Table 2: Write Height instead of Hight

Response 3: Thanks for your suggestion. This is a very basic mistake, in the template writing, there is an automatic line break which causes each line to contain a coherent word, this error was not identified in subsequent revisions, we have made revisions to all the errors in the manuscript that occurred with this situation.

Thanks again for your valuable suggestion.

#comment-4:

  1. In Discussion: Line no. 247 to 248 - Reference should be given for the statement, Line no. 287: Write between instead of be-tween

Response 4: Thanks for your suggestion. We have added the reference source for this statement, please refer to reference 29 for details.

Thanks again for your valuable suggestion.

Reviewer 2 Report

Comments and Suggestions for Authors

The manuscript (MS) deals with regeneration processes in a conifer mountain forest. In principle, the research design is quite transparent and the results may be of interest for the readership engaged into forest conservation and restoration.

Some issues should be clarified and corrected.

Scientifically, there are a couple of issues.

l. 135-155 -> the authors speak of the saplings' sizes as dependent on elevation. However, nothing is said on the age of the saplings which may be a stronger determinant of the sizes--as the plants grow with time. This point should be clarified.

l. 193-200: 'RDA was also used to rank the contribution rates of soil parameters to soil variability at each altitude (Figure 5).' -> However, the caption of Figure 5 is 'Contribution rates of soil stoichiometry parameters at each elevation to renewal indices', which is an obvious mismatch. This point should be clarified and corrected.

l. 248-262, 296 -> on the one hand, the authors speak of apoplastic decomposition, on the other, of apomictic decomposition. This should be clarified. Also, 'apoplastic decomposition' seems to be a bit unusual term. Probably, what is meant is decomposition of dead xylem? 'Apoplastic' is usually used in a couple with 'symplastic' as transport routes in plants.

Minor comments:

l. 14, 52, 92-94, 235 -> the Latin names should be given with the authors' names at the first appearance in the text.

l. 14: a ecologically -> an ecologically

Table 1. -> because soil thickness, aspect, and slope are not the results of a single measurement, shouldn't they be given with variations like other parameters?

l. 334: carbon to nitrogen ratio(C/P) -> probably, carbon to phosphorus ratio?

Comments on the Quality of English Language

English is quite clear and above average for non-native speakers, as I can judge. Still, slight corrections might be required.

Author Response

#comment-1:

135-155 -> the authors speak of the saplings' sizes as dependent on elevation. However, nothing is said on the age of the saplings which may be a stronger determinant of the sizes--as the plants grow with time. This point should be clarified.

Response 1: Thanks for your suggestion. In field investigations, determining the age of trees poses challenges, particularly when it comes to seedlings. Instead of age structure, research commonly employs diameter class structure (diameter at breast height) to analyze the structural characteristics of populations across different diameter classes. Our study did not specifically address the structural characteristics of Qinghai spruce seedlings. The primary objective of this study is to examine the relationship between density, diameter at breast height, tree height, and stoichiometric parameters of Qinghai spruce regeneration indicators. So, the age structure of Qinghai spruce seedlings was not scrutinized in this study, this is also an important direction for our future research and your suggestions are very important for my future research. I will focus on this aspect of the research.

Thanks again for your valuable suggestion.

#comment-2:

193-200: 'RDA was also used to rank the contribution rates of soil parameters to soil variability at each altitude (Figure 5).' -> However, the caption of Figure 5 is 'Contribution rates of soil stoichiometry parameters at each elevation to renewal indices', which is an obvious mismatch. This point should be clarified and corrected.

Response 2: Thanks for your suggestion. we are sorry for the inaccurate description of Figure 5 and we have revised, the updated caption is "Analysis of soil chemical stoichiometric parameter variability at different elevations based on redundancy analysis (RDA) model" to the manuscript in line 232-233.

Thanks for your suggestion.

#comment-3:

248-262, 296 -> on the one hand, the authors speak of apoplastic decomposition, on the other, of apomictic decomposition. This should be clarified. Also, 'apoplastic decomposition' seems to be a bit unusual term. Probably, what is meant is decomposition of dead xylem? 'Apoplastic' is usually used in a couple with 'symplastic' as transport routes in plants.

Response 3:Thank you for your suggestion. we are sorry for unprofessional description caused your confusion, and we have made the necessary revisions to the manuscript in lines 272 - 291. The term "Apoplastic" used was also inaccurate, and we have corrected it to "humus".

Thanks for your suggestion.

Minor comments:

14, 52, 92-94, 235 -> the Latin names should be given with the authors' names at the first appearance in the text.

14: a ecologically -> an ecologically

Response1: Thank you for your suggestion. We have checked all Latin names of plant and made revisions.

Thanks for your suggestion.

Table 1. -> because soil thickness, aspect, and slope are not the results of a single measurement, shouldn't they be given with variations like other parameters?

Response2: Thank you for your suggestion. We have reorganized all the original data and made corrections to the manuscript in Table 1.

Thanks for your suggestion.

334: carbon to nitrogen ratio(C/P) -> probably, carbon to phosphorus ratio?

Response3: Thank you for your suggestion. This is another basic mistake.

Thanks for your suggestion.

Reviewer 3 Report

Comments and Suggestions for Authors

Dear authors, thank you for conducting such an interesting manuscript. Below, I present to you my suggestions for improvement of the manuscript.

Introduction - I would be very interested in how these forests are managed. What type of management, what intensity, some spatial management information. Maybe it would be more suitable for Materials & Methods, but it has definitely also important impacts on soil characteristics. 2–3 sentences with citations should be dedicated to this.

You also write that your results will be important for conservation and management, but again, what is the current state of protection and management of the Pices crassifolia mountain stands?

If it is a National Park, why would be your results important if the area already enjoys maximum protection?

Please, improve and extend the Introduction.

I do not see any clear results, please highlight in Conclusion the top 2–3 results and what are they implications for protection and management as stated in aims of the study.

Please, add some photos of regen and stands as supplementary material.

Comments on the Quality of English Language

I recommend one more check by native English speaker.

Author Response

#comment-1:

Introduction - I would be very interested in how these forests are managed. What type of management, what intensity, some spatial management information. Maybe it would be more suitable for Materials & Methods, but it has definitely also important impacts on soil characteristics. 2–3 sentences with citations should be dedicated to this.

You also write that your results will be important for conservation and management, but again, what is the current state of protection and management of the Pices crassifolia mountain stands?

If it is a National Park, why would be your results important if the area already enjoys maximum protection?

Response 1: Thanks for your suggestion. In recent years, Qilian Mountain Forest have already enjoys maximum protection, with no human destruction, the primary human activity in the area is grazing. We have improved and extended the introduction, introduced the situation of forest management in the Qilian Mountains Forest to the manuscript in lines 63-68, and introduced forest management in the experimental design to the manuscript in lines 119-121.

Thanks again for your valuable suggestion.

#comment-2:

I do not see any clear results, please highlight in Conclusion the top 2–3 results and what are they implications for protection and management as stated in aims of the study.

Response 2: Thanks for your suggestion. We have carefully summarized and emphasized the conclusion once again, in order to provide a clearer presentation to the manuscript in line 347-368.

Thanks again for your valuable suggestion.

#comment-3:

Please, add some photos of regen and stands as supplementary material.

Response 3: Thanks for your suggestion. we added photos of regeneration seedings and stands in the introduction as shown Figure 1.

Thanks again for your valuable suggestion.
